# Association of Statin Therapy with Functional Outcomes and Survival in Intracerebral and Subarachnoid Hemorrhage

**DOI:** 10.3390/neurolint17020027

**Published:** 2025-02-10

**Authors:** Bahadar S. Srichawla, Daksha Gopal, Majaz Moonis

**Affiliations:** Department of Neurology, University of Massachusetts Chan Medical School, Worcester, MA 01655, USA; daksha.gopal@umassmed.edu (D.G.); majaz.moonis@umassmed.edu (M.M.)

**Keywords:** intracerebral hemorrhage, subarachnoid hemorrhage, stroke, statins, HMG-CoA reductase

## Abstract

**Background/Objectives**: Intracerebral hemorrhage (ICH) and subarachnoid hemorrhage (SAH) are severe forms of stroke with high morbidity and mortality rates. HMG-CoA reductase inhibitors, commonly referred to as statins, known for their lipid-lowering abilities, also possess pleiotropic properties, including anti-inflammatory and neuroprotective effects. We aimed to evaluate the impact of statin therapy on the functional outcomes and survival in patients with ICH and SAH. **Methods**: This retrospective cohort study analyzed data from the Get With The Guidelines (GWTG) stroke registry at a tertiary care center, including patients diagnosed with ICH or SAH between January 2008 and June 2022. Patients were categorized based on prior initiation of statin therapy: no statin, low-intensity statin, or high-intensity statin. The primary outcome was the Modified Rankin Scale (mRS) score at discharge, dichotomized to good (0–2) and poor (3–6) outcomes. A multivariate logistic regression model controlled for age, gender, and National Institutes of Health Stroke Scale (NIHSS) score at admission. **Results**: A total of 663 patients with ICH and 159 patients with SAH were included in the analysis. In the ICH patients, low-intensity statin therapy was associated with significantly higher odds of a good functional outcome (aOR 2.56, 95% CI 1.247–5.246, *p* = 0.0104), as was high-intensity statin therapy (aOR 2.445, 95% CI 1.313–4.552, *p* = 0.0048). Among the SAH patients, all 39 deaths occurred in the no statin therapy group. **Conclusions**: Both low- and high-intensity statin therapy are associated with improved functional outcomes in ICH and may offer a survival benefit in SAH. These findings highlight the potential neuroprotective role of statins in hemorrhagic stroke. Further prospective studies and randomized controlled trials are needed to confirm these observations and to clarify the optimal use of statins in this patient population.

## 1. Introduction

Intracerebral hemorrhage (ICH) and subarachnoid hemorrhage (SAH) are severe forms of stroke with high morbidity and mortality. Despite advances in medical care, the prognosis for patients with these types of hemorrhagic stroke remains poor, with many survivors experiencing significant long-term functional impairments [1]. Identifying therapies that can improve both survival and functional outcomes in these patients is a critical area of research.

HMG-CoA reductase inhibitors, commonly referred to as statins, are widely known for their cholesterol-lowering properties and have been extensively studied in the context of cardiovascular disease prevention [2,3]. However, beyond their lipid-lowering effects, statins exhibit pleiotropic properties, including anti-inflammatory, neuroprotective, and endothelial-stabilizing effects, which may confer benefits in the context of cerebrovascular diseases [4]. These properties have led researchers to investigate the potential role of statins in improving outcomes in various forms of stroke, including ischemic stroke. However, their role in hemorrhagic stroke, particularly ICH and SAH, remains less well-defined.

Recent studies have suggested that statin therapy may be associated with improved outcomes in patients with hemorrhagic stroke, potentially by modulating the inflammatory response, reducing secondary brain injury, and enhancing vascular repair mechanisms [5]. Despite these promising findings, the evidence remains mixed, and concerns about the safety of statin use in the acute setting of hemorrhagic stroke, particularly regarding the risk of rebleeding, persist [6]. This study aims to explore the relationship between statin use and functional outcomes, as well as survival rates, in patients who have experienced ICH or SAH.

## 2. Materials and Methods

### 2.1. Study Population and Primary Endpoints

Patients diagnosed with ICH or SAH between January 2008 and June 2022 at our single tertiary care center in Massachusetts were included. Data were obtained from the Get With The Guidelines (GWTG) stroke registry for our tertiary care center. The GWTG stroke registry was developed in part to improve outcomes in patients diagnosed with an acute stroke [7]. The inclusion criteria were (i) patients age between 18 and 89 years and (ii) with computerized tomography (CT) or magnetic resonance imaging (MRI) confirming the presence of either ICH or SAH as assessed by a board-certified neurologist. The exclusion criteria included patients with an ischemic stroke, transient ischemic attack, or other neurovascular pathologies. Patients’ age, gender, laboratory data (low-density lipoprotein [LDL], high-density lipoprotein [HDL]) vascular risk factors, pre-admission medications, National Institutes of Health Stroke Scale (NIHSS) score at admission, and Modified Rankin Scale (mRS) score at discharge were manually abstracted from the electronic medical records. The study population was divided into those not on statin therapy, low-intensity statin therapy, and high-intensity statin therapy. Low-intensity statins were defined as those that lower LDL-C by less than 30%, and high-intensity statins as those that lower LDL-C by 50% or more [8,9]. The primary endpoint was the discharge mRS score as well as mortality.

### 2.2. Statistical Analysis

Categorical variables are expressed as frequencies and percentages. Quantitative variables are reported as the mean (standard deviation, SD) or median (25th–75th percentile), as appropriate. For the main statistical analysis, we dichotomized the mRS score into good (0–2) and poor (3–6). An mRS score of 6 was categorized as deceased and was used to determine mortality. We created a multivariate logistic regression model to determine whether statin therapy (both high and low intensity) was independently associated with disability and mortality in patients with ICH. The multivariate model controlled for age, gender, and NIHSS score at admission. Due to the limited sample, size only mortality data were assessed in the SAH group. All the statistical analyses were completed using SAS Institute^®^ SAS^®^ (SAS Institute^®^—Cary, NC, USA).

### 2.3. Data Availability and Considerations

The investigators will share the anonymized data (with the associated coding library) used in developing the results presented in this research upon reasonable request from investigators who have received appropriate ethical clearance from their host institution. This research was conducted following the Declaration of Helsinki, developed by the World Medical Association, on the ethical principles for medical research involving human subjects. This study was approved by the institutional review board (IRB) of the University of Massachusetts Chan Medical (Approval Code: STUDY00002066, Approval Date: 10 March 2024).

## 3. Results

### 3.1. Intracerebral Hemorrhage and Statins

A total of 1235 patients with ICH were identified from the GWTG registry. However, 572 records were excluded due to the lack of mRS (n = 327) and NIHSS (n = 245) scores. A total of 663 patients had data available on statin therapy and the post-discharge mRS score. Here, 95 patients were on high-dose statin therapy, 76 patients on low-dose statin therapy, and 492 patients not on statin therapy. A flow chart of the patients is included in Figure 1. The clinical characteristics of the included ICH patients are provided in Table 1. The study population was stratified based on the statin dosage into three groups: high-dose statins (n = 95), low-dose statins (n = 76), and no statin use (n = 492). The gender distribution varied significantly across the groups, with males constituting 62.1% (59) of the high-dose group, 59.2% (45) of the low-dose group, and 48.8% (240) of the no statins group, resulting in a total male representation of 51.9% (344) (*p* = 0.0234). Conversely, females accounted for 37.9% (36) of the high-dose group, 40.8% (31) of the low-dose group, and 51.2% (252) of the no statins group, contributing to an overall female distribution of 48.1% (319). A multivariate logistic regression model controlling for age, gender, and NIHSS score at admission is presented in Table 2. Low-intensity statin therapy was associated with significantly higher odds of having a good outcome (mRS 0–2) compared to those not on statin therapy (aOR 2.557 CI 1.247–5.246, *p* = 0.0104). High-intensity statin therapy was also significantly associated with higher odds of a good outcome (aOR 2.445 CI 1.313–4.552, *p* = 0.0048). When comparing high-intensity statins to low-intensity statins, we found the odds of a good outcome were not statistically higher (aOR 0.958, CI 0.438–2.096 *p* = 0.9154).

### 3.2. Subarachnoid Hemorrhage and Statins

A total of 534 patients with SAH were identified from the GWTG registry. However, 375 records were excluded due to missing mRS (n *=* 148) and NIHSS (n *=* 227) scores. Here, 19 patients were on high-dose statin therapy, 16 patients on low-dose statin therapy, and 124 patients not on statin therapy. A flow chart of the patients is included in Figure 2. The clinical characteristics of the included SAH patients are provided in Table 3. A total of 39 deaths were reported (39/159; 24.5%), all of which occurred in the no stain group. No patients were reported to have died who were on statin therapy and diagnosed with SAH. The study population of SAH patients was categorized into three groups based on the statin usage: high-dose statins, low-dose statins, and no statins. Among the high-dose statin group, males accounted for 31.6% (6) of the population, while females comprised 68.4% (13). In the low-dose statin group, males made up 43.8% (7) of the population, and females constituted 56.3% (9). The no statin group showed a slightly different distribution, with males representing 42.7% (53) and females accounting for 57.3% (71). Overall, across all the groups, the total male and female distributions were 41.5% (66) and 58.5% (93), respectively, with a *p*-value of 0.6433, indicating no statistically significant difference in the gender distribution among the groups.

## 4. Discussion

This study aimed to evaluate the impact of statin therapy on the functional outcomes and survival in patients with ICH and SAH, two forms of hemorrhagic stroke associated with high morbidity and mortality. Our findings suggest that statin therapy, particularly high-intensity statins, is associated with better functional outcomes in ICH patients and may also play a protective role in the survival of SAH patients. Our study revealed that both low-intensity and high-intensity statin therapy were significantly associated with improved functional outcomes in ICH patients, as indicated by the higher likelihood of achieving a good mRS score of 0–2 at discharge. These results are consistent with previous research indicating that statins may exert neuroprotective effects that extend beyond their lipid-lowering capabilities [10]. The pleiotropic effects of statins, including the anti-inflammatory and endothelial-stabilizing properties, may contribute to the observed benefits in ICH patients [11]. The neuroprotective mechanisms of statins in the context of hemorrhagic stroke may include the reduction of secondary brain injury through the attenuation of inflammatory responses, stabilization of the blood–brain barrier, and promotion of endothelial repair [12]. In particular, the inhibition of HMG-CoA reductase by statins may lead to a decrease in the production of pro-inflammatory cytokines and reactive oxygen species, both of which are implicated in the pathogenesis of secondary brain injury following ICH [13]. Furthermore, the enhancement of endothelial function by statins could improve the cerebral blood flow and reduce the risk of further hemorrhage, thereby contributing to better functional recovery [14].

Sturgeon et al. investigated the risk factors for intracerebral hemorrhage (ICH) in a pooled cohort from the ARIC and CHS studies, involving over 21,000 participants followed for stroke events. Over 263,489 person-years, 135 ICH cases were identified, with age, African American ethnicity, and hypertension being significant risk factors, while lower LDL cholesterol and triglycerides were inversely associated with the ICH risk. Factors like sex, smoking, alcohol intake, BMI, and diabetes were not significantly associated with ICH, highlighting the critical role of blood pressure management in reducing the ICH risk [15]. Noda et al. examined the relationship between low LDL cholesterol levels and the risk of death from intraparenchymal hemorrhage in a cohort of over 91,000 adults aged 40–79 years, who were followed for 10 years. They found that individuals with LDL cholesterol ≥ 140 mg/dL had about half the risk of death from intraparenchymal hemorrhage compared to those with LDL cholesterol < 80 mg/dL, with a consistent inverse relationship even after adjusting for cardiovascular risk factors and conducting sensitivity analyses. These findings suggest that low LDL cholesterol levels are associated with an increased risk of death from intraparenchymal hemorrhage [16]. Wang et al. completed a meta-analysis of 23 prospective studies involving 1,430,141 participants and 7960 hemorrhagic strokes to assess the relationship between cholesterol levels and the hemorrhagic stroke risk. They found that higher total cholesterol and LDL cholesterol levels were inversely associated with the risk of hemorrhagic stroke, with a pooled relative risk of 0.69 for high versus low total cholesterol and 0.62 for LDL cholesterol. In contrast, higher HDL cholesterol levels appeared to increase the risk of intracerebral hemorrhage, highlighting the distinct associations between different cholesterol subtypes and hemorrhagic stroke [17].

Although our study was limited by the small sample size in the SAH cohort, the absence of deaths in the statin-treated group is noteworthy. This finding suggests a potential protective effect of statin therapy on survival in SAH patients, which warrants further investigation. However, careful consideration should be given to a selection bias, which may have influenced our findings. Previous studies have reported mixed results regarding the impact of statins on SAH outcomes, with some suggesting benefits in terms of reduced vasospasm and delayed cerebral ischemia, while others have raised concerns about potential adverse effects [18,19]. The lack of mortality in the statin-treated SAH patients in our cohort may be related to the anti-inflammatory and vasoprotective effects of statins, which could mitigate the severity of the initial hemorrhage and subsequent complications, such as cerebral vasospasm. The lack of temporal data concerning when statin therapy was initiated prior to the acute brain injury may also influence the outcomes, which was not captured in our dataset. Additionally, statins have been shown to enhance nitric oxide bioavailability, which may contribute to improved cerebral vasodilation and perfusion, potentially reducing the risk of ischemic events following SAH [20,21].

This study has several limitations that should be acknowledged. First, the retrospective nature of this study introduces the possibility of selection bias, and the exclusion of patients with missing data may have impacted the generalizability of our findings. Second, the sample size in the SAH cohort, particularly among the statin-treated patients, was small, limiting the statistical power to detect significant associations. The observational design of this study precludes the establishment of a causal relationship between statin therapy and improved outcomes. Although the statin intensity was abstracted, data on specific agents were not available. Other limitations include the lack of meaningful disease-specific clinical outcomes within this database, including cerebral vasospasm, recurrent bleeds, etc. The location of the ICH (e.g., lobar, subcortical), cause of the ICH, and the mechanism of the SAH (e.g., traumatic, aneurysmal, convexal) were not factors that could be assessed in this retrospective study. Future directions should involve high-quality multi-center randomized controlled trials that follow international consensus guidelines for the management of these acute brain injuries. These studies should control for confounding comorbidities. Additionally, studies examining the optimal timing, dosing, and duration of statin therapy in the acute setting of hemorrhagic stroke are needed to inform clinical guidelines. Differentiating the lipid-lowering effects of statins and their pleiotropic effects influencing hemorrhagic stroke should also be considered [22]. Understanding the potential risks and benefits of statin therapy in this context is crucial for optimizing patient outcomes and improving the management of hemorrhagic stroke.

## 5. Conclusions

In this study, we explored the association of statin therapy with the functional outcomes and survival in patients with intracerebral hemorrhage (ICH) and subarachnoid hemorrhage (SAH). Our findings suggest that statin therapy (both low and high intensity) is associated with improved functional outcomes in patients with ICH. Furthermore, although limited by the sample size, our data indicate a potential survival benefit for patients with SAH who are on statin therapy. However, due to this study’s observational nature and the limitations in terms of the sample size, particularly in the SAH cohort, caution is warranted when interpreting these findings.

## Figures and Tables

**Figure 1 neurolint-17-00027-f001:**
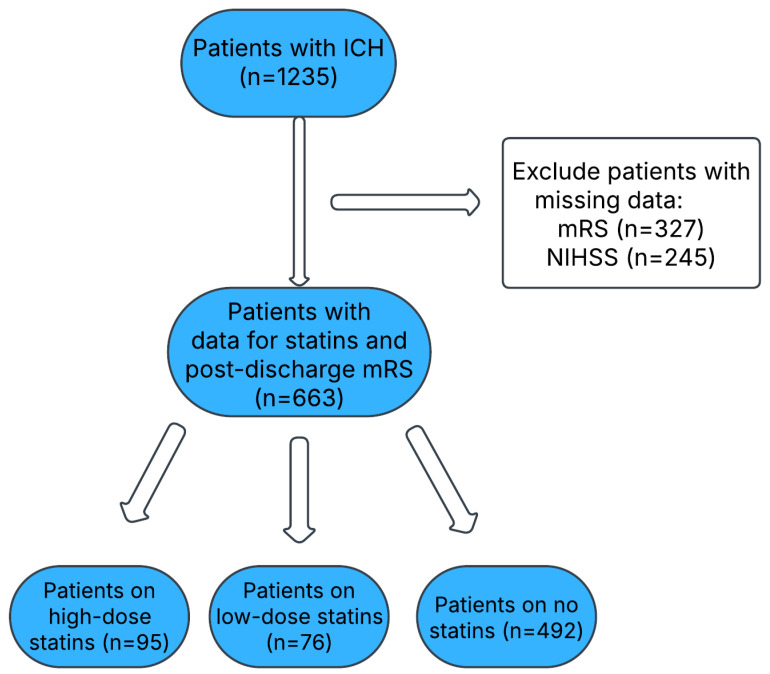
Flow diagram of the included ICH patients.

**Figure 2 neurolint-17-00027-f002:**
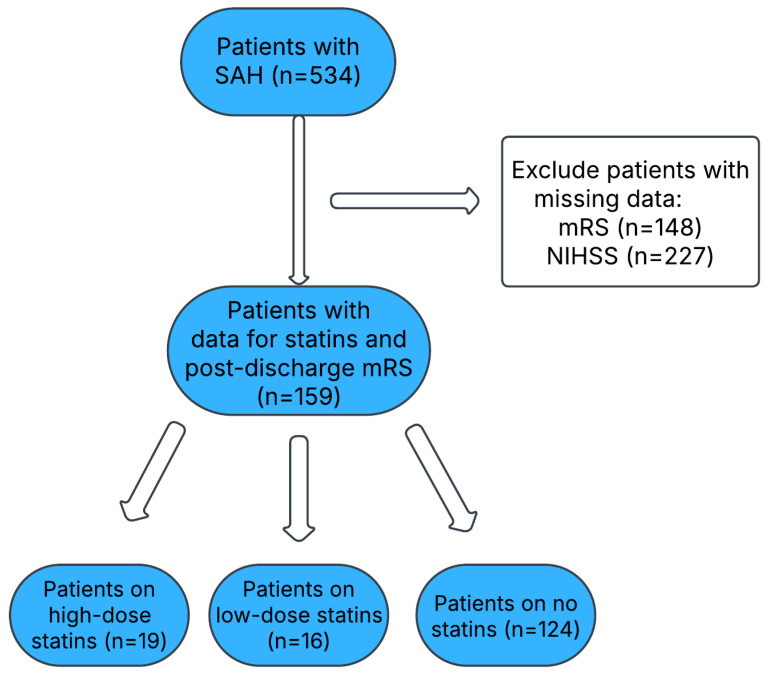
Flow diagram of the included SAH patients.

**Table 1 neurolint-17-00027-t001:** Descriptive characteristics of the included ICH patients.

Clinical Characteristics	Category	High Dose (N = 95)	Low Dose (N = 76)	No Statins (N = 492)	Total	*p*-Value
Gender	Male	62.1 (59)	59.2 (45)	48.8 (240)	51.9 (344)	0.0234 *
	Female	37.9 (36)	40.8 (31)	51.2 (252)	48.1 (319)	
Age	<45	3.2 (3)	1.3 (1)	6.3 (31)	5.3 (35)	0.1242
	45–65	35.8 (34)	25 (19)	27.8 (137)	28.7 (190)	
	65+	61.1 (58)	73.7 (56)	65.9 (324)	66.1 (438)	
BMI	Underweight	3 (2)	4.7 (2)	1.6 (3)	2.4 (7)	0.7719
	Normal	24.2 (16)	25.6 (11)	28.7 (54)	27.3 (81)	
	Overweight	33.3 (22)	34.9 (15)	38.3 (72)	36.7 (109)	
	Obese	39.4 (26)	34.9 (15)	31.4 (59)	33.7 (100)	
Atrial fibrillation	Yes	27.4 (26)	23.7 (18)	10.6 (52)	14.5 (96)	<0.0001 *
	No/ND	72.6 (69)	76.3 (58)	89.4 (440)	85.5 (567)	
Diabetes mellitus	Yes	30.5 (29)	19.7 (15)	2 (10)	8.1 (54)	<0.0001 *
	No/ND	69.5 (66)	80.3 (61)	98 (482)	91.9 (609)	
No medical history	Yes	2.1 (2)	0 (0)	4.3 (21)	3.5 (23)	0.1226
	No/ND	97.9 (93)	100 (76)	95.7 (471)	96.5 (640)	
Anti-hypertensive medication usage	Yes	77.9 (74)	78.9 (60)	59.8 (294)	64.6 (428)	<0.0001 *
	No/ND	22.1 (21)	21.1 (16)	40.2 (198)	35.4 (235)	
Diabetes mellitus medication usage	Yes	29.5 (28)	18.4 (14)	7.1 (35)	11.6 (77)	<0.0001 *
	No/ND	70.5 (67)	81.6 (62)	92.9 (457)	88.4 (586)	
Cholesterol-reducing medication	Yes	80 (76)	84.2 (64)	32.5 (160)	45.2 (300)	<0.0001 *
	No/ND	20 (19)	15.8 (12)	67.5 (332)	54.8 (363)	
Antidepressant usage	Yes	14.7 (14)	26.3 (20)	5.9 (29)	9.5 (63)	<0.0001 *
	No/ND	85.3 (81)	73.7 (56)	94.1 (463)	90.5 (600)	
Anticoagulant usage	Yes	4.2 (4)	9.2 (7)	3.5 (17)	4.2 (28)	0.0675
	No/ND	95.8 (91)	90.8 (69)	96.5 (475)	95.8 (635)	
Antiplatelet usage	Yes	10.5 (10)	15.8 (12)	7.1 (35)	8.6 (57)	0.0329 *
	No/ND	89.5 (85)	84.2 (64)	92.9 (457)	91.4 (606)	
Other antithrombotic medication usage	Yes	71.6 (68)	67.1 (51)	47.4 (233)	53.1 (352)	<0.0001 *
	No/ND	28.4 (27)	32.9 (25)	52.6 (259)	46.9 (311)	
No prior medication usage	Yes	6.3 (6)	3.9 (3)	10.6 (52)	9.2 (61)	0.1023
	No/ND	93.7 (89)	96.1 (73)	89.4 (440)	90.8 (602)	
MRI	No = 0	78.9 (75)	89.5 (68)	89.2 (439)	87.8 (582)	0.0176 *
	Yes = 1	21.1 (20)	10.5 (8)	10.8 (53)	12.2 (81)	
Pre-stroke mRS	1	78.3 (47)	80 (28)	77.5 (110)	78.1 (185)	0.9688
	2	15 (9)	11.4 (4)	13.4 (19)	13.5 (32)	
	3	6.7 (4)	8.6 (3)	9.2 (13)	8.4 (20)	
NIHSS performed	Yes	100 (95)	100 (76)	100 (492)	100 (663)	
Grouped discharge mRS	0–2	26.3 (25)	22.4 (17)	11.2 (55)	14.6 (97)	<0.0001 *
	3–5	73.7 (70)	77.6 (59)	48.4 (238)	55.4 (367)	
	6	0 (0)	0 (0)	40.4 (199)	30 (199)	
Discharge mRS	0	15.8 (15)	9.2 (7)	5.9 (29)	7.7 (51)	<0.0001 *
	1	4.2 (4)	5.3 (4)	4.1 (20)	4.2 (28)	
	2	6.3 (6)	7.9 (6)	1.2 (6)	2.7 (18)	
	3	7.4 (7)	3.9 (3)	3.7 (18)	4.2 (28)	
	4	55.8 (53)	55.3 (42)	29.7 (146)	36.3 (241)	
	5	10.5 (10)	18.4 (14)	15 (74)	14.8 (98)	
	6	0 (0)	0 (0)	40.4 (199)	30 (199)	
Statin usage	N	0 (0)	0 (0)	100 (492)	74.2 (492)	<0.0001 *
	Y	100 (95)	100 (76)	0 (0)	25.8 (171)	
Mortality	No	100 (95)	100 (76)	59.6 (293)	70 (464)	<0.0001 *
	Yes	0 (0)	0 (0)	40.4 (199)	30 (199)	

The * represents statistical significance.

**Table 2 neurolint-17-00027-t002:** Multivariate logistic regression analysis of the ICH patients.

Effect	OR	95% Wald		*p*-Value
		Confidence Limits		
High-dose statin vs. no statin	2.445	1.313	4.552	0.0048
High-dose statin vs. low-dose statin	0.958	0.438	2.096	0.9154
Low-dose statin vs. no statin	2.557	1.247	5.246	0.0104
Gender	0.902	0.536	1.519	0.699
Age	0.935	0.917	0.952	<0.0001
NIHSS admission	0.857	0.818	0.897	<0.0001

Additionally adjusted for age, gender, and NIHSS score at admission.

**Table 3 neurolint-17-00027-t003:** Descriptive characteristics of the included SAH patients.

Clinical Characteristics	Category	High-Dose Statin (N = 19)	Low-Dose Statin (N = 16)	No Statin (N = 124)	Total	*p*-Value
Gender	M	31.6 (6)	43.8 (7)	42.7 (53)	41.5 (66)	0.6433
	F	68.4 (13)	56.3 (9)	57.3 (71)	58.5 (93)	
Age	<45	5.3 (1)	0 (0)	12.1 (15)	10.1 (16)	0.2147
	45–65	31.6 (6)	31.3 (5)	42.7 (53)	40.3 (64)	
	65+	63.2 (12)	68.8 (11)	45.2 (56)	49.7 (79)	
BMI	Underweight	0 (0)	0 (0)	7 (3)	4.8 (3)	0.1869
	Normal	11.1 (1)	10 (1)	27.9 (12)	22.6 (14)	
	Overweight	33.3 (3)	70 (7)	27.9 (12)	35.5 (22)	
	Obese	55.6 (5)	20 (2)	37.2 (16)	37.1 (23)	
Atrial fibrillation	Yes	26.3 (5)	6.3 (1)	5.6 (7)	8.2 (13)	0.0088 *
	No/ND	73.7 (14)	93.8 (15)	94.4 (117)	91.8 (146)	
Diabetes mellitus	Yes	0 (0)	12.5 (2)	3.2 (4)	3.8 (6)	0.1223
	No/ND	100 (19)	87.5 (14)	96.8 (120)	96.2 (153)	
No medical history	Yes	5.3 (1)	0 (0)	6.5 (8)	5.7 (9)	0.5738
	No/ND	94.7 (18)	100 (16)	93.5 (116)	94.3 (150)	
Anti-hypertensive medication usage	Yes	73.7 (14)	62.5 (10)	47.6 (59)	52.2 (83)	0.0722
	No/ND	26.3 (5)	37.5 (6)	52.4 (65)	47.8 (76)	
Diabetes mellitus medication usage	Yes	0 (0)	6.3 (1)	3.2 (4)	3.1 (5)	0.5695
	No/ND	100 (19)	93.8 (15)	96.8 (120)	96.9 (154)	
Cholesterol-reducing medication	Yes	78.9 (15)	93.8 (15)	25.8 (32)	39 (62)	<0.0001 *
	No/ND	21.1 (4)	6.3 (1)	74.2 (92)	61 (97)	
Antidepressant usage	Yes	10.5 (2)	6.3 (1)	4 (5)	5 (8)	0.4702
	No/ND	89.5 (17)	93.8 (15)	96 (119)	95 (151)	
Anticoagulant usage	Yes	5.3 (1)	0 (0)	0 (0)	0.6 (1)	0.0245 *
	No/ND	94.7 (18)	100 (16)	100 (124)	99.4 (158)	
Antiplatelet usage	Yes	10.5 (2)	12.5 (2)	0.8 (1)	3.1 (5)	0.006 *
	No/ND	89.5 (17)	87.5 (14)	99.2 (123)	96.9 (154)	
Other antithrombotic medication usage	Yes	73.7 (14)	56.3 (9)	28.2 (35)	36.5 (58)	0.0001 *
	No/ND	26.3 (5)	43.8 (7)	71.8 (89)	63.5 (101)	
No prior medication usage	Yes	5.3 (1)	0 (0)	15.3 (19)	12.6 (20)	0.1303
	No/ND	94.7 (18)	100 (16)	84.7 (105)	87.4 (139)	
MRI	No = 0	94.7 (18)	81.3 (13)	91.1 (113)	90.6 (144)	0.3573
	Yes = 1	5.3 (1)	18.8 (3)	8.9 (11)	9.4 (15)	
Pre-stroke mRS	1	100 (9)	87.5 (7)	75 (27)	81.1 (43)	0.3541
	2	0 (0)	12.5 (1)	8.3 (3)	7.5 (4)	
	3	0 (0)	0 (0)	16.7 (6)	11.3 (6)	
NIHSS performed	Yes	100 (19)	100 (16)	100 (124)	100 (159)	
Grouped discharge mRS	0–2	47.4 (9)	31.3 (5)	35.5 (44)	36.5 (58)	0.0016 *
	3–5	52.6 (10)	68.8 (11)	33.1 (41)	39 (62)	
	6	0 (0)	0 (0)	31.5 (39)	24.5 (39)	
Discharge mRS	0	42.1 (8)	18.8 (3)	22.6 (28)	24.5 (39)	0.0045 *
	1	5.3 (1)	6.3 (1)	9.7 (12)	8.8 (14)	
	2	0 (0)	6.3 (1)	3.2 (4)	3.1 (5)	
	3	10.5 (2)	0 (0)	2.4 (3)	3.1 (5)	
	4	21.1 (4)	56.3 (9)	21.8 (27)	25.2 (40)	
	5	21.1 (4)	12.5 (2)	8.9 (11)	10.7 (17)	
	6	0 (0)	0 (0)	31.5 (39)	24.5 (39)	
Statin usage	N	0 (0)	0 (0)	100 (124)	78 (124)	<0.0001 *
	Y	100 (19)	100 (16)	0 (0)	22 (35)	
Mortality	0	100 (19)	100 (16)	68.5 (85)	75.5 (120)	0.0007 *
	1	0 (0)	0 (0)	31.5 (39)	24.5 (39)	

The * represents statistical significance.

## Data Availability

Data can be made available upon reasonable request from the Editor-in-Chief to the corresponding author.

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
