# Peer review of "Association of Statin Therapy with Functional Outcomes and Survival in Intracerebral and Subarachnoid Hemorrhage"

_2035-8377, 2025, doi:10.3390/neurolint17020027_

Round 1
Reviewer 1 Report
Comments and Suggestions for Authors
The authors of the paper entitled, 'Impact of statin therapy on functional outcomes and survival in intracerebral and subarachnoid hemorrhage' have tried to investigate the use of statins vis-a-vis functional outcomes in two severe stroke forms. The paper is well-written; however, the authors are suggested to clarify the following points for better comprehension.
1. The authors should mention in the introduction section that why only mRS is used as functional outcome other than mortality in both ICH and SAH whereas, other important variables such as hematoma expansion, cerebral vasospasm, re-bleeding, stroke-specific quality of life and cognitive outcomes are overlooked.
2. The authors must write elaborate definitions of low intensity and high intensity statin dosing according to the latest guidelines on the treatment of blood cholesterol to reduce cerebrovascular or atherosclerotic cardiovascular risk in adults by the American College of Cardiology/American Heart Association Task Force. The definitions of low intensity statins and high intensity statins provided by the authors are not convincing.
3. The authors must give explanations that which type of statins (atorvastatin, rosuvastatin, simvastatin, pravastatin, lovastatin) were used for the treatment of ICH or SAH individually as several statins have different impacts on disease-stage, severity and outcome.
4. The authors must mention the quantity of dose as low intensity, moderate intensity, or high intensity for better clarity.
5. The authors should clarify that why the time duration of the statin use with respect to the date of hospitalization is not taken care of because longer duration of statin use has a different impact than shorter durations, which would have substantiated the impact of low intensity and high intensity statin use in functional outcome of ICH and SAH.
6. The authors should mention that why triglyceride levels were not taken from the medical records of the patients, an important parameter significantly associated with ICH irrespective of HDL and LDL levels.
7. Re-write line 81.
8. Lines 85-86: Are bad mRS scores classified as (0-6)? See its terminology also, in the abstract section, these are mentioned as good and poor, here it is mentioned as bad score.
9. The age range i.e. 18-89 years is somewhat confounding as younger individuals are less likely of being prescribed statins, which would have induced false positivity in the results. All patients (n=39) who died were those not using statins. Are these patients who died were younger in age or due to selection bias? This must be discussed in the discussion section.
10. According to the results in the present study, low intensity of statins provide better functional outcomes than high-dose intensity (ORs 2.557 vs. 2.445), if it is true then why the authors are advocating high intensity of statins as a better option for functional outcomes especially for SAH mortality?
11. The results and conclusion of the present study are based on the comparisons of high-dose statin vs. no statin and low-dose statin vs. no statin, which narrates half of the story. There must be another comparison i.e. high-dose statin vs. low-dose statin. This group comparison would have substantiated the results of whether low-intensity or high-intensity statins are better for functional outcomes in ICH and SAH. By the way, a dose of statin is different from intensity (defined by the authors in the methods section) which the authors must take care of while designing or interpreting the results.
Author Response
Comments 1: The authors should mention in the introduction section that why only mRS is used as functional outcome other than mortality in both ICH and SAH whereas, other important variables such as hematoma expansion, cerebral vasospasm, re-bleeding, stroke-specific quality of life and cognitive outcomes are overlooked.
Response 1: Thanks for brining up this important point. We have edited the limitations section to include this as a limitation with GWTG database. Granular details regarding cerebral vasospasm, and subarachnoid hemorrhage specific outcomes were not curated in this database. (Line 174-176 page 11).
Comment 2: The authors must write elaborate definitions of low intensity and high intensity statin dosing according to the latest guidelines on the treatment of blood cholesterol to reduce cerebrovascular or atherosclerotic cardiovascular risk in adults by the American College of Cardiology/American Heart Association Task Force. The definitions of low intensity statins and high intensity statins provided by the authors are not convincing.
Response 2: Thanks for brining up this important point. The definitions provided in the methods section of the manuscript (page 2, line 77-80) highlight the definitions of low-intensity and high-intensity statin therapy. As requested we added references by the ASA/AHA to support this point.
Comment 3: The authors must give explanations that which type of statins (atorvastatin, rosuvastatin, simvastatin, pravastatin, lovastatin) were used for the treatment of ICH or SAH individually as several statins have different impacts on disease-stage, severity and outcome.
Response 3: Unfortunately the specific statin therapy used was not included in the GWTG stroke database only the intensity of the statin. This has also been added to the limitations section of the manuscript (line 173 page 11).
Comment 4: The authors should clarify that why the time duration of the statin use with respect to the date of hospitalization is not taken care of because longer duration of statin use has a different impact than shorter durations, which would have substantiated the impact of low intensity and high intensity statin use in functional outcome of ICH and SAH.
Response 4: Thanks for brining up this important point. Indeed the stroke database utilized all us to assess for statin usage prior to hospitalization but not the exact time-frame prior to the neurological insult. The authors agree this temporal pattern could influence outcomes and this has been made clear to the readership in the limitations section of the manuscript. (page 11; lines 179-180).
Comment 5: The authors must mention the quantity of dose as low intensity, moderate intensity, or high intensity for better clarity.
Response 5: Thanks, this has been clarified on (page 2, line 77-80)
Comment 6: The authors should mention that why triglyceride levels were not taken from the medical records of the patients, an important parameter significantly associated with ICH irrespective of HDL and LDL levels.
Response 6: Thanks for bringing up this point. The authors agree triglycerides level may play an important role in subarachnoid hemorrhage and hemorrhagic management. This laboratory data was not captured from this dataset. Likewise, current practice guidelines recommend aggressive control of LDL in patients with ischemic stroke. Additionally, this manuscript aims to elucidate on the potential role of LDL and intracerebral and subarachnoid hemorrhage which is currently an area of significant interest. This is of particular interest to the field of vascular neurology as there is conflicting evidence if low LDL may exacerbate hemorrhagic stroke. This reference has been included within the study. (PMID:35159988)
Comment 7: Re-write line 81.
Response 8: Completed. "The primary endpoint was the discharge mRS as well as mortality".
Comment 8: Lines 85-86: Are bad mRS scores classified as (0-6)? See its terminology also, in the abstract section, these are mentioned as good and poor, here it is mentioned as bad score.
Response 8: The authors apologize this was a typo. mRS scores of 3-6 was considered poor as stated in the abstract. This has been rectified. Thank you.
Comment 9: The age range i.e. 18-89 years is somewhat confounding as younger individuals are less likely of being prescribed statins, which would have induced false positivity in the results. All patients (n=39) who died were those not using statins. Are these patients who died were younger in age or due to selection bias? This must be discussed in the discussion section.
Response 9: The demographic breakdown including the number of patients who were not on statins and there age range is included in Table 3. This table shows that those who were age <45 (n=15), 45-65 years (n=53), and age >65 (n=56). As the SAH group showed that mortality was only in the non-statin group a statistical analysis is not possible as there was not a single death reported in either the low- or high-intensity statin groups. Therefore, this finding was presented to the reader. Selection bias must always be taken into consideration and we further elaborated on this in the discussion section. Lines 171-173; page 11.
Comment 10: According to the results in the present study, low intensity of statins provide better functional outcomes than high-dose intensity (ORs 2.557 vs. 2.445), if it is true then why the authors are advocating high intensity of statins as a better option for functional outcomes especially for SAH mortality?
Response 10: Thanks for brining up this important point. Indeed the OR was higher for the low-intensity statin group however, the p value was lower in high-intensity statin group with smaller confidence intervals (p = 0.0048 vs. p = 0.0104). However, we have edited the abstract and conclusions section of the manuscript to reflect the reviewers points particularly with regards to SAH.
Comment 11: The results and conclusion of the present study are based on the comparisons of high-dose statin vs. no statin and low-dose statin vs. no statin, which narrates half of the story. There must be another comparison i.e. high-dose statin vs. low-dose statin. This group comparison would have substantiated the results of whether low-intensity or high-intensity statins are better for functional outcomes in ICH and SAH. By the way, a dose of statin is different from intensity (defined by the authors in the methods section) which the authors must take care of while designing or interpreting the results.
Response 11: Thanks for bringing up this important point. The authors have completed an additional multivariate analysis on high-intensity vs. low-intensity statin therapies and we report that when comparing high-intensity statins to low-intensity statins we found the odds of a good outcome was not statistically higher (aOR 0.958, CI 0.438 – 2.096 p = 0.9154). This information has been added to Table 2 as well as results section page 3 line 119-121.
Reviewer 2 Report
Comments and Suggestions for Authors
This is an interesting and well-done study. It presents a design that is not optimal, but nevertheless, it adequately addresses the topic for the available resources.
The title is synthetic, clear and appropriate. However, I suggest changing the word impact to association, since the study cannot study impact or cause-effect relationships, but only the existence or not of associations that must be studied, analyzed and confirmed.
The summary is appropriate, brief, synthetic, presents data on quantifiable and adequately measured results, and describes unbiased conclusions in an objective manner. It is adequately divided into sections.
The introduction is quite brief, but it is sufficient to substantiate the problem and raise the question and objectives of the study.
The methodology is adequately described with sufficient detail to guarantee the reproducibility of the study. As mentioned above, the type of design is not ideal, but it is adequate for the available resources.
The results are presented in an objective, clear and unbiased manner. High-quality tables and graphs are used.
The discussion presents some objectionable comments. In principle, it would be appropriate to improve the description of the limitations by including: (1) emphasizing the need for a multicenter design, (2) highlighting that a single-center registry linked to a project to comply with international guidelines is used, therefore, it shows that these are patients registered for potentially not complying with the recommendation guidelines, (3) the need to quantify the population base that the hospital serves, in order to consider the real public health impact of these results, (4) lack of information on other data linked to the use or "non-use" of statins. For example: would it be possible that patients who did not use statins had fewer health checks and treatment of other pathologies such as arterial hypertension and/or health screening? Could the statin be just an epiphenomenon not linked to the change in evolution?
Finally, in conclusions it is stated that: "In this study, we explored the impact of statin therapy." In that sentence, "impact" should be replaced by "possible association."
The sentence "These results underscore the potential neuroprotective effects...." should be deleted, since it is not a conclusion, but an abstract theoretical argument/postulation.
Author Response
Comments 1: However, I suggest changing the word impact to association, since the study cannot study impact or cause-effect relationships, but only the existence or not of associations that must be studied, analyzed and confirmed.
Response 1: The title has been changed as requested.
Comments 2: (1) emphasizing the need for a multicenter design, (2) highlighting that a single-center registry linked to a project to comply with international guidelines is used, therefore, it shows that these are patients registered for potentially not complying with the recommendation guidelines, (3) the need to quantify the population base that the hospital serves, in order to consider the real public health impact of these results, (4) lack of information on other data linked to the use or "non-use" of statins. For example: would it be possible that patients who did not use statins had fewer health checks and treatment of other pathologies such as arterial hypertension and/or health screening? Could the statin be just an epiphenomenon not linked to the change in evolution?
Response 2: Thanks for bringing up this important point. We have highlighted and included this as a future directions section at page 11; line 188-192.
Comments 3: Finally, in conclusions it is stated that: "In this study, we explored the impact of statin therapy." In that sentence, "impact" should be replaced by "possible association."
Response 3: Completed.
Comments 4: The sentence "These results underscore the potential neuroprotective effects...." should be deleted, since it is not a conclusion, but an abstract theoretical argument/postulation.
Response 4: Completed.
Reviewer 3 Report
Comments and Suggestions for Authors
The topic of this paper is actual and interesting as the effect of statin therapy in subarachnoid hemorrhage and intracerebral hemorrhage remains to be clarified. Reports in the literature are non so consistent and conclusive. The study is retrospective and a first criticism is represented by the large part of cases excluded from the study due to lack of data (NIHSS and mRS). This should represent a potential bias. A second aspect is that groups of patient are really different in size. No information about reason for statin therapy and low-high dose administration. Despite this aspects, the work suggest a potential benefit from statin therapy in hemorrhagic stroke patients and should be a further base for future research. The paper is clear and well written. Methods are clearly described, results are also clear and support conclusion. Discussion in essential. I consider the paper should bene considered for the publication in the present form.
Author Response
Comments 1: The study is retrospective and a first criticism is represented by the large part of cases excluded from the study due to lack of data (NIHSS and mRS). This should represent a potential bias. A second aspect is that groups of patient are really different in size. No information about reason for statin therapy and low-high dose administration.
Response 1: Thanks for brining up these important points we have highlighted them as potential limitations and to be taken for consideration by the readership. Page 11.
Reviewer 4 Report
Comments and Suggestions for Authors
Dear Authors,
I had the pleasure of reading your manuscript. There are some issues that must be improved.
1. It is not clear in "Methods" which are the outcomes of the study and when were them assessed.
2. The authors must define what they considered as low dose statins and high dose statins.
3. It is not clear if the authors also included cases of SAH.
4. Do the authors have any information on the etiology of the hemorrhage? If not this must be included as a limitation.
5. The authors did not assess adherence to statin therapy. It is proved it can influence the outcomes. This must be included as a limitation.
6. The authors did not monitor the lipid profile to compare the effect of the cholesterol reduction itself or the pleiotropic effect of the statin therapy on functional outcomes. This must be included as a limitation, citing and discussing doi: 10.1080/01616412.2021.1967677.
Author Response
Comments 1: It is not clear in "Methods" which are the outcomes of the study and when were them assessed.
Response 1: The outcomes of the study were mortality as well as discharge modified Rankin Score (mRS) this was stated on page 2 line 80.
Comments 2: The authors must define what they considered as low dose statins and high dose statins.
Response 2: Thank you this has been added to the methods section (page 2, line 76-79).
Comments 3: It is not clear if the authors also included cases of SAH.
Response 3: Yes, SAH patients are included in the study as stated in the title, abstract, methods, the results section highlight a flow chart for included SAH patient. As stated in the results a regression analysis was not completed for the SAH group as no patients within the statin group died.
Comments 4: Do the authors have any information on the etiology of the hemorrhage? If not this must be included as a limitation.
Response 4: Unfortunately, etiology of the hemorrhage was not included in the database and this has been added as a limitation.
Comment 5: The authors did not assess adherence to statin therapy. It is proved it can influence the outcomes. This must be included as a limitation.
Response 5: All patients included within the analysis were on statin therapy. There were no patients included within the statin therapy group which were not taking the medication. Thanks.
Comment 6:The authors did not monitor the lipid profile to compare the effect of the cholesterol reduction itself or the pleiotropic effect of the statin therapy on functional outcomes. This must be included as a limitation, citing and discussing doi: 10.1080/01616412.2021.1967677.
Response 6: Thanks for bringing up such a valuable point this has been included in the limitations section for the readership and cited within the limitations/future directions section of the manuscript. (page 12, line 225-227).
Round 2
Reviewer 1 Report
Comments and Suggestions for Authors
I think all is well in the end, regarding comment no. 18, let me explain my opinion, both ORs are significant according to P values. The degree of significance can differentiate only when ORs are the same, but if OR is higher for something, that value should be taken. So don't hesitate to write in the manuscript, that low-intensity statin therapy is better in your study.
Author Response
Comments 1: I think all is well in the end, regarding comment no. 18, let me explain my opinion, both ORs are significant according to P values. The degree of significance can differentiate only when ORs are the same, but if OR is higher for something, that value should be taken. So don't hesitate to write in the manuscript, that low-intensity statin therapy is better in your study.
Response 1:
Thanks for your recommendation, we have edited the abstract and conclusions of the manuscript to highlight then low-intensity statin therapy may be more beneficial and requires future research.